# Characterization of New Small-Spored *Alternaria* Species Isolated from *Solanaceae* in Algeria

**DOI:** 10.3390/life11121291

**Published:** 2021-11-24

**Authors:** Nabahat Bessadat, Bruno Hamon, Nelly Bataillé-Simoneau, Kihal Mabrouk, Philippe Simoneau

**Affiliations:** 1UMR IRHS, University Angers, Agrocampus-Ouest, INRAE, SFR QUASAV, 49071 Beaucouzé, France; bessadat.nabahat@univ-oran1.dz (N.B.); bruno.hamon@univ-angers.fr (B.H.); nelly.bataille-simoneau@univ-angers.fr (N.B.-S.); 2Laboratoire de Microbiologie Appliquée, Université Oran1 Ahmed Ben Bella, BP1524 El M’naouer, Oran 31000, Algeria; kihal.mebrouk@univ-oran1.dz

**Keywords:** small-spored, *Alternaria*, *Solanaceae*, taxonomy, diversity

## Abstract

Although large-spored *Alternaria* species of the section *Porri* are considered to be the major agents responsible for leaf spot and blight of *Solanaceae*, small-spored *Alternaria* species are also frequently isolated from symptomatic tissues. A survey of the north-western regions of Algeria during the 2017–2018 growing seasons revealed that amongst the 623 *Alternaria* isolates from tomato, potato, pepper, eggplant and black nightshade, 8% could not be morphologically assigned to either section *Porri* or section *Alternaria*. In order to more precisely determine the taxonomic position of these isolates, detailed morphological characterizations and multi-locus phylogenetic analyses were performed. Based on these analyses, the isolates were grouped into four main clades: section *Ulocladioides*, section *Infectoriae*, including two new species, section *Embellisioides*, and section *Eureka*, including one new species. These isolates were also characterized for their virulence under green-house conditions. They were able to produce leaf spot symptoms on tomato plants but with variable levels.

## 1. Introduction

*Alternaria* foliar diseases are important factors that decrease yield in solanaceous crops [1,2]. Of these foliar diseases, early blight and brown spot are very common [3,4]. Early blight starts with the formation of small, brownish spots on lower leaves that enlarge and develop concentric rings. Leaves turn yellow and plants may defoliate. Brown spot also initially develops on lower leaves with dark-brown lesions but the foliar lesions never develop the dark concentric rings; instead they coalesce across large veins until whole leaves turn brown and hang from the plant. The pathogens spread progressively to other plants or parts of the same plant such as the stems, fruits and tubers, where they may initiate new infections [5,6,7,8,9].

Many *Alternaria* species have been documented to cause these foliar diseases on *Solanaceae*. Among large-spored species belonging to the section *Porri*, *A. solani* has long been recognized as the major species responsible for early blight [10,11]. However, it has been shown that *A. linariae*, *A. grandis* and *A. protenta* can also infect both tomato (*Solanum lycopersicum* L.) and potato (*Solanum tuberosum* L.) [12,13,14,15]. Moreover, several small-spored species have been reported to cause necrotic lesions on *Solanaceae*, e.g., *A. alternata*, *A. arborescens*, *A. tenuissima*, *A. dumosa,*
*A. interrupta* [8,14,16,17,18,19]. However, the exact number of species responsible for *Solanaceae* leaf spot diseases should be taken with caution due to the recent redefinition of species within the genus *Alternaria*, based on multi-gene phylogeny. For instance, Woudenberg et al. [20] synonymized some of the small-spored species (*A. tenuissima*, *A. dumosa* and *A. interrupta*) under *A. alternata* and showed that members of the *A. arborescens* species constitute a species complex. Nevertheless, concerning tomato, members of at least seven out of the 28 sections described within the genus *Alternaria* [21,22] have been associated with foliar diseases. Besides the large-spored species of the section *Porri* and the members of the section *Alternaria* (*A. alternata*, *A. arborescens, A. tomato*), this includes the following species: *A. mimicula* (section *Brassicicola*) [23], *A. consortialis* (section *Ulocladioides*) [14], *A. infectoria* (section *Infectoriae*) [24], *A. chlamydosporigena* (section *Embellisia*) [25] and more recently *A. telliensis* (section *Japonicae*) [26].

*Solanaceae*, such as potato and tomato, are the most important crops and the second main source of food in Algeria, after cereals. Annual surveys across the north-western regions of the country revealed the presence of a high rate of *Alternaria* diseases on several plant species. In fact, early blight is considered to be a major threat for *Solanaceae* production since *A. grandis* and *A. protenta* were observed on tomato and *A. linariae* on potato for the first time [27,28,29]. The north-western regions of Algeria are characterized by a warm Mediterranean climate (Köppen climate classification) where tomato can be cultivated throughout the year [30]. In these regions, other solanaceous cultivated plants, i.e., potato, eggplant (*Solanum melongena*), and pepper (*Capsicum annuum*), are often found in the near vicinity of tomato fields or even used in crop rotation with tomato. Climatic conditions and agricultural practices are therefore highly favorable for the development of epidemics and the appearance of new virulent genotypes. Thus, the composition of *Alternaria* species that is associated with tomato foliar diseases is probably rapidly evolving in Algeria and may be more complex than previously recognized. Moreover, except for the members of the section *Alternaria* [18,31], small-spored *Alternaria* spp. have been poorly studied and were the primary focus of this study. The objective was thus to estimate the diversity of small-spored species associated with necrotic lesions on *Solanaceae* in Algeria using morphological, multi-locus phylogenetic and pathological approaches. This study revealed that small-spored isolates belonging to four sections of *Porri,* other than section *Alternaria,* were isolated from symptomatic tissues of *Solanaceae* in northern Algeria.

## 2. Materials and Methods

### 2.1. Sampling and Isolation

The current study was part of a long-term project on the species diversity of the *Solanaceae–Alternaria* pathosystem in the north-west region of Algeria [14]. During the 2017 and 2018 growing seasons, samples with typical symptoms of early blight and brown spot were collected from several solanaceous plants in the cropping area of the Mostaganem regions. Weeds such as black nightshade (*Solanum nigrum*) and jimson weed (*Datura stramonium*) with suspected foliar symptoms of *Alternaria* infection were also collected (Table 1). Samples were separately placed in paper bags and transported to the laboratory. The isolation of fungi was performed immediately. Small pieces (1 cm^2^) of both diseased and apparently healthy vegetable tissue (leaf, stem, fruit) were collected, and their surfaces were disinfected by dipping them in 2% sodium hypochlorite solution for 2 min and rinsing them with sterile distilled water three times. Samples were placed in Petri plates containing potato carrot agar medium (PCA) and incubated at room temperature in ambient light. Multiple fungal colonies that were growing up from the margins or surfaces of each lesion produced conidia within one to three weeks. These colonies were examined under a light stereomicroscope at 20× and 40× and further transferred onto new PCA plates. The colonies with *Alternaria* characteristics were purified by transferring single spores onto fresh potato dextrose agar medium (PDA) plates. After adequate growth, mycelial plugs of these isolates were cut and stored in 30% glycerol at −80 °C to maintain a fungal collection.

### 2.2. Species Identification and Morphological Characterization

Pure cultures of *Alternaria* isolates with different morphologies and representing small-spored species were preliminarily identified according to the standard method [23]. Isolates were cultured on 9 cm PCA plates and incubated at 22–28 °C under a daily fluorescent 8 light/16 dark cycle for 5 to 7 days. After this period, the arrangement of conidia on conidiophores and the patterns of chain branching were examined at 40× and 100× without disrupting the colony. For the examination of microscopic details, mounts were prepared in pure lactic acid without added dye. The conidial color, shape, size (length and width) and number of septa (longitudinal, oblique or transverse), as well as the presence or absence of secondary conidiophores were recorded from observations that were made at 400X magnifications. Dimensions are based on the observation of 50 conidia and 25 conidiophores per isolate. In general, sporulation patterns were compared with the representative species described in the literature (Simmons 2004; 2007) or with the reference strain of *A. cumini* (CBS 121329). For macro-morphological characterization, the color of the upper and reverse side of the colonies, colony margin, texture and diameter were recorded on PCA and PDA. Color notations were rated according to the color charts of Kornerup and Wanscher [32]. New species were also described on PDA, Oatmeal Agar (OA) and Malt Extract Agar (MEA). The effect of temperature on growth was studied at different temperatures, i.e., 4 °C, 25 °C, 30 °C, 35 °C and 40 °C. Petri dishes (Ø = 90 mm) containing 15 mL of each media (PCA, PDA, MEA, OA) were inoculated with 5 mm mycelial discs of 15 days-old culture. Colony diameter was measured after 7 days of incubation. Specimen for each new species (NB65, NB530, NB560, NB562, NB568, NB660) were deposited in the Westerdijk Fungal Biodiversity Institute (Utrecht, The Netherlands) under the reference codes CBS 146567, CBS 148413, CBS 148415, CBS 148414, CBS 148416, and CBS 146566, respectively. Nomenclatural data were deposited in MycoBank (MB 835037: *A. cuminicola*, MB 840829: *A. dahraensis*, MB 840845: *A. pseudohumuli*).

### 2.3. DNA Extraction, Polymerase Chain Reaction (PCR) Amplification and Sequencing

*Alternaria* isolates were grown on PDA for 7 days; mycelium was scraped from the culture surface and transferred into a microtube. Lysis buffer (50 mM Tris-HCl pH 7.5, 50 mM EDTA, 3% SDS, 1% 2-mercaptoethanol) was added and the nucleic acids were extracted according to the microwave mini-prep procedure described by Goodwin and Lee [33]. The final DNA pellet was dissolved into 100 μL TE buffer (10 mM Tris-HCl pH 8.0, 0.1 mM EDTA).

As an initial step of identification, the internal transcribed spacer region of ribosomal DNA (*ITS rDNA*) and a portion of the glyceraldehyde-3-phosphate dehydrogenase (*gpd*) gene were amplified from the isolates with the primers ITS1/ITS4 [34] and gpd1-gpd2 [35], respectively. Since it was impossible to discriminate between closely related species solely by using the *ITS rDNA* and *gpd* sequences, portions of the genes of the translation elongation factor 1-alpha (*tef1*), RNA polymerase’s second largest subunit (*rpb2*), plasma membrane ATPase (*ATPase*), actin (*act*), a major *Alternaria* allergen (*Alt a1*), and a mating type protein (MAT1-2-1) were amplified with the primer pairs EF1-728F/EF1-986R [36], RPB2–5F2/fRPB2–7cR [37,38], ATPDF1/ATPDR1, ACTDF1/ACTDR1 [39], Alt-for/Alt-rev [40] and MAT1-2F/MAT1-2R [41], respectively. PCR amplification was carried out in a total volume of 50 μL containing 75 mM Tris-HCl pH 9.0, 20 mM (NH_4_)_2_SO_4_, 0.01% (*w*/*v*) Tween 20, 1.5 mM MgCl_2_, 200 μM of each deoxyribonucleotide triphosphate, 1 unit of thermostable DNA polymerase (GoTaq, Promega) and 400 nM of each relevant oligonucleotide primer. The thermocycling parameters were the same as described in the references provided above. DNA amplification products were analyzed by electrophoresis in 1.2% (*w*/*v*) agarose in 0.5 × TAE buffer and sent for sequencing at GATC Lab (Germany).

### 2.4. Sequence Alignment and Phylogenetic Analysis

DNA sequences were concatenated and aligned by the MUSCLE algorithm using MEGA 7 [42]. Phylogenetic analysis was performed using the maximum likelihood (ML) and Bayesian inference (BI) approaches under IQTree v.1.6. [43] and MrBayes v.3.2.1 [44], respectively. The best-fit evolutionary models for each dataset were calculated by ModelFinder [45] under the Bayesian Information Criterion (BIC) selection procedure. The ML analysis was carried out with 1,000 ultrafast bootstrap replicates and only values above 70% were considered significant. BIs were performed in order to estimate the posterior probabilities (PP) of tree topologies based on the Markov Chain Monte Carlo (MCMC) analysis with four chains, 1M generations, sampling intervals of 1,000 generations. Burn-in was set to 25% and only PP values above 0.95 were considered significant.

### 2.5. Pathogenicity Tests

In order to determine the virulence of the isolates, representative isolates selected from among the identified phylogenetic taxa were used (Table 1). Conidia of these isolates that were produced on 7 to 10 day-old PCA cultures were used to prepare conidial suspensions. Conidia were collected by rubbing them gently from the surface of the medium with a rubber spatula and placing them in a test tube with sterile distilled water containing 0.01% Tween 80; the conidial suspensions were then filtered through a mousseline membrane to remove mycelial fragments. Concentrations of the spore suspensions were estimated using a hemocytometer and adjusted to 10^5^ conidia per mL. Tomato var. Saint Pierre and potato var. Désirée were used in this study. Disinfected tomato seeds and potato tubers were grown, after germination, in plastic pots containing sterilized soil (3/4 potting soil, 1/4 sand mixture). Six *Apiaceae* (coriander, carrot var. Super Muscade, parsley, fennel, celery, cumin) plant species were included in the pathogenicity tests in order to assess the host range of some isolates from section *Eureka*. Three plants from each species were used in this study. The plants were kept in a greenhouse for six weeks. Greenhouse temperatures were approximately 25–32 °C during the day and 19–25 °C at night. Inoculation was done by spraying the plants with 10 mL of the spore suspension. Five strains, three from section *Alternaria* (*A. alternata*, *A. arborescens*), one from section *Eureka* (*A. cumini*, CBS 121329) and one from section *Radicina* (*A. petroselini* CBS 109383) were used as positive controls on tomato and *Apiaceae* plants, respectively. The negative controls were sprayed with sterile distilled water. Plants were covered with polyethylene plastic bags for two days to maintain a high level of humidity. The percentage of the leaves’ surfaces that were covered with dark lesions was estimated for each plant 7, 14 and 21 days after inoculation (DAI). Each inoculation experiment was done in triplicate.

## 3. Results

### 3.1. General Survey

A large survey was carried out in 2017 and 2018 in the Mostaganem region (Algeria) on symptomatic leaves of different cultivated solanaceous crops (tomato, eggplants, potato, pepper) or neighboring wild species belonging to the *Solanaceae* (jimson weed *Datura stramonium*, black nightshade *Solanum nigrum*). A total of 367 leaf samples representing brown spot and early blight symptoms were collected, and 1061 isolates were obtained and observed for their conidia size and morphology. Based on these observations, the isolated fungi were classified into four categories: small-spored *Alternaria* spp. (48%), large-spored *Alternaria* spp. (11%), *Stemphyllium* spp. (33%) and fungi belonging to other genera (8%). Among the small-spored *Alternaria* species, 92% had typical sporulation patterns of *Alternaria* sect. *Alternaria* and were present in all sampling locations, while 8% could not be unambiguously identified at the section level based only on their morphological characteristics. A selection of 20 small-spored isolates that produced conidia that were different from the isolates of the members of the section *Alternaria* were subjected to monospore purification for further characterization (Table 1). The initial characterization of isolates from this collection was performed by sequencing the *ITS rDNA* and *gpd* loci. The latter locus allowed a good resolution of the 28 sections described in the genus *Alternaria.* The resulting phylogenetic tree (Figure 1) revealed that the isolates belonged to four sections: *Ulocladioides*, *Infectoriae*, *Embellisioides* and *Eureka*.

### 3.2. Species Identification

Further identification of these isolates was obtained through multilocus sequence analyses and detailed morphological characterization.

#### 3.2.1. Section Ulocladioides

The two isolates assigned to this section (NB558, NB559) shared similar morphological characteristics. They both had a slow radial growth rate (ca. 8.67 ± 0.89 mm/day on PDA medium) and formed dark green, powdery colonies. Microscopic observations (Figure 2a,b) showed that they produced obovoid, beakless, small-size conidia (mean length and width: (15)25–36(41) µm and (8)10–13(15) µm, respectively), with 1–5 transverse septa and 0–4 longitudinal septa. Conidiophores arose near the agar surface and were (25)30–50(77) µm long and 3–4(5) µm wide and had 1–13 successive, apical sympodial conidiogenous sites. Secondary conidiophores were scarcely formed. A multilocus phylogeny based on *gpd, Alt a1* and *MAT1-2-1*, which has been previously used to resolve the phylogenetic relationships between species that were formerly included in the genus *Ulocladium* [48], was selected to identify the two isolates at the species level. Sequences obtained for NB558 and NB559 at these 3 loci were compared with those from 16 of the 21 recognized species belonging to section *Ulocladioides* [48,49], for which the sequences were available in GenBank. Sequences of an isolate of *A. alternata* (CBS 916.96) were used as the outgroup. The combined gene dataset included 1,849 characters (including gaps) and 135 were parsimony-informative. The best-fit model (TN+F+I+G4) was selected for the analysis. The topology of the resulting trees that were inferred from the two phylogenetic methods (ML and BI) was similar. The ML phylogenetic tree shown in Figure 3 did not allow for the complete separation of all of the species within section *Ulocladioides,* and the isolates NB508 and NB509 could not be separated from two species, i.e., *A. cucurbitae* and *A. brassicae-pekinensis.* Additional attempts to resolve this group of strains by including two additional loci (*rpb2* and *tef1*) were unsuccessful (Appendix A).

#### 3.2.2. Section Infectoriae

Twelve of the small-spored isolates considered here were included in the section *Infectoriae* based on *gpd* phylogeny (Figure 1). They all shared morphological characteristics that are typical for members of this section, i.e., they often have mostly mycelial colonies in the culture, with poor sporulation until hyphae were scarified, conidiophores with several conidiogenous loci, resulting in a three-dimensional sporulation pattern, and many conidia with elongated apical regions (“false beaks”) [50]. Primary conidiophores were mostly simple or branched ((12)30–75(130) × 3–4(5) µm) and had 1–4 conidiogenous loci. Conidia were obclavate, ellipsoid to ovoid and reached a size of (15)25–44(72) × (6)8–12(17) µm with 2–11 transverse septa and 0-5 longitudinal septa (Figure 2c–g). However, due to the high similarity between isolates, it was not possible to delineate reliable morpho-groups based solely on morphology within the members of this section. Phylogenetic analyses at five loci (*ITS rDNA*, *gpd*, *rpb2*, *tef1* and *ATPase*) have been used to separate species within the section *Infectoriae* [51,52,53,54,55]. Sequences were obtained at these five loci for the 12 isolates assigned to section *Infectoriae* and compared with corresponding sequences from 37 recognized species from section *Infectoriae*. Sequences of *A. abundans* CBS 534.83 (section *Chalastospora*) were used as the outgroup. The combined dataset included 2,566 characters (including gaps) and 193 were parsimony-informative. The best-fit evolutionary model was TN+F+R4. The resulting ML tree confirmed that all of the isolates clustered in section *Infectoriae* (Figure 4). However, most of the isolates appeared as lineages distinct from the 37 types of strains of the species included in this study and independently formed separated branches (e.g., NB561, NB540, NB539, NB538, NB550). Isolates NB562, NB530 and NB545 clustered together, forming a highly supported clade that was distinct from all the other strains included in that analysis. Similarly, NB560 and NB568 grouped in a clade together with *A. humuli*. However, the phylogenetic relationship between these two isolates and this species was uncertain due to the low support of internal branches. Based on these observations, we propose two new species in the section *Infectoriae* named *Alternaria dahraensis* and *Alternaria pseudohumuli*, respectively (see description below in the “Taxonomy” paragraph).

#### 3.2.3. Sections Embellisioides and Eureka

As shown in Figure 1, six isolates, all collected from the Mostaganem region, grouped into two phylogenetically related sections, i.e., section *Embellisioides* and section *Eureka*, that both contain species formerly assigned to the genus *Embellisia* [21,46]. Five isolates (NB65, NB542, NB557, NB660, NB666), clustering with members of the section *Eureka* based on *gpd* phylogeny, shared similar morphology. These isolates developed greyish green, velvety colonies and produced short, beakless, ovoid conidia ((12)25–42(52) × (6)10–20(25) µm). The conidia were formed on long primary conidiophores ((155)430–760(1097) × 3-5(6) µm). These isolates usually produced short conidial chains (3–4 units) and secondary conidiophores ((17)25–55(158) × 3–5 µm), which arose from apical, lateral or basal cells of the conidia. The tertiary conidiophores were common and produced single conidia. Sporulation patterns were almost the same compared with the *A. cumini* reference strain CBS 212,329, with the exception of the conidia size ((30)35–50(70) × (12)15–17(21) µm) and the abundance of tertiary conidiophores. Isolate NB354 had different morphological characteristics (Figure 2h) and produced ellipsoid, ovoid, conical or cylindrical conidia with broadly rounded bases (mean length and width (30)40–48(61) × (10)12–15(25) µm, respectively). Conidia were solitary (rarely in short chains of 2–3 spores) or produced in a bouquet, with mostly transverse septa (4–8) that distinctly contrasted with the external wall. Conidiophores were of moderate length, (32)45–57(89) µm, each bearing 1–8(14) conidiogenous sites. These morphological characteristics were in agreement with the assignation of this isolate to the section *Embellisioides.* In order to better resolve these isolates at the species level within their respective sections, a phylogenetic analysis based on additional loci was performed. In accordance with Woudenberg et al. [46], who first resolved the taxonomic position of strains that were formerly grouped in the genus *Embellisia*, a combined *gpd-rpb2-tef1* phylogeny was used. Sequences were obtained at these three loci for the six isolates that were assigned to sections *Eureka* and *Embellisioides* and compared with the corresponding sequences from the 12 recognized species from these sections. Sequences of *A. embellisia* CBS 339.71 (section *Embellisia*) were used as the outgroup. The combined dataset included 1,533 characters (including gaps) and 217 were parsimony-informative. The best-fit evolutionary model was TNe+G4. The resulting ML tree (Figure 5) revealed that isolate NB354 formed a well-supported branch close to *A. lolii* within section *Embellisioides.* As the morphology of this isolate matched the published description of *Embellisia lolii* from E.G. Simmons & C.F. Hill [56], isolate NB354 was considered to be a member of this species. By contrast, the five isolates belonging to section *Eureka* formed a separate clade that was closely related to *A. cumini*. Comparison of sequences from these isolates with those from the reference strain of *A. cumini* at additional loci (*Act, ATPase, Alt a1*) confirmed that they formed a distinct cluster (Appendix A). As described above, discrete morphological characteristics distinguished these isolates from *A. cumini* and they were therefore considered to be a new species named *Alternaria cuminicola* (see detailed description below in the “Taxonomy” paragraph).

#### 3.2.4. Taxonomy

*Alternaria dahraensis N. Bessadat*, and *P. Simoneau* sp. nov. (Figure 6)MycoBank: MB 840829

Etymology: name refers to the Dahra range located in the northern part of Algeria where the fungus was isolated.

Culture ex-type specimen: NB 530 = CBS 148413Other specimen examined: NB545, NB 562 = CBS 148414

Description: Colonies on PCA are subhyaline to brown, arachnoid to loosely wooly, approximately 76 mm diameter at day 7 (Figure 6i); abundant sporulation but not densely crowded at the aerial hyphae within the 3 cm central radius of the NB350 colony. However, colonies of strain NB562 were mostly mycelial, with a meager sporulation. Zones of growth and sporulation were not well defined. Aerial arachnoid hypha arose early over the center of the colony and gradually over the entire colony; they mainly produced scattered chains of conidia. Conidia chains that were viewed at 40X had a conspicuous appearance of open-angled tufts of 7–15(34) conidia. Chains were simple in young colonies (3–6 conidia) (Figure 6k); they became branched with open angles to the primary chain upon aging (Figure 6m). These chains included 1–2(3) lateral branches of 1–3(5) conidia. Primary conidiophores that arose near the agar surface were 57 ± 25 × 5 ± 1 (15–113 × 4–6) µm in size, with 3–8(18) transverse septa, were usually simple, occasionally branched, and incorporated 1–3 conidiogenous loci. Secondary conidiophores were usually short and often (but not always) the dominant element in the chain structure; they were frequently longer than conidium body (Figure 6k). These lateral or apical secondary conidiophores (Figure 6b,e) were single-celled; they had either straight, geniculate, or angular extensions *ca* 56 ± 25 × 5 ± 1 (5–90(118) × 4–7) µm, with 1 to 3 conidiogenous loci. The apical or lateral secondary conidiophore of the primary conidia was as long as the primary conidiophore and became conidiogenous. Simple secondary conidiophores of exceptional length were often present within most of aerial spore clusters. Conidia were ovoid or ellipsoid, medium brown, smooth or ornamented, and each was beakless with either a bluntly tapered or conical apical cell (Figure 6a,d). Longitudinal septa were common within the total maturing population. The largest conidia in the mature colony were mostly basal in chains near 39 ± 10 × 11 ± 2 (18–68 × 8–23) µm with 3–8(10) transverse septa and 1–4 longitudinal septa, although some short, ovoid, primary conidia were present. Most secondary conidia were of the same length or shorter *ca* 36 ± 6 × 14 ± 2 (23–53 × 9–18) µm. A large number of conidia remained small within the chains (Figure 6f); they were commonly in the range of 29 ± 7 × 14 ± 2 (19–45 × 10–20) µm, with 1–4(6) transverse septa and 0–4 longitudinal septa.

*Alternaria pseudohumuli N. Bessadat*, and *P. Simoneau* sp. nov. (Figure 7)MycoBank: MB 840845Etymology: name refers to the closely related species *A. humuli*.Culture ex-type specimen: NB 560 = CBS 148415Other specimen examined: NB 568 = CBS 148416

Description: the colonies on PCA were often mostly mycelial, hyaline to subhyaline, loosely wooly, and reached 66 mm diameter after 7 days (Figure 7j); sporulation was meager or completely lacking until hypha were disturbed or scarified. Zones of growth and sporulation were not well defined. Clusters of a few conidia (up to 5 to 15) were already scattered on the aerial hyphae above the center of the colony after 4 days. By day 14, discrete clusters of openly branched conidial chains were produced on the agar substrate near the center of the colony, resulting in bushy clumps (Figure 7d,i). Primary conidiophores were solitary, straight or geniculate, brown, simple, occasionally branched, short *ca* 25–82 × 3–5 (53 ± 15 × 4 ± 1) μm with 1–7 transverse septa; some were long (with up to 8 transverse septa) *ca* 63–133 × 3–7 (95 ± 18 × 5 ± 0.5) μm, with one or several (2–6) conidiogenous loci (pores) at the apex and throughout their lengths (Figure 7i,l). Conidial chains were moderately long to long 4–7(13) and usually branched. Branching was produced by the occurrence of several conidiogenous loci on apical secondary conidiophores (Figure 7b,e) and lateral secondary conidiophores (Figure 7h); lateral branches of 2–3(5) conidia were present and the dominant portion of the sporulation pattern (in contrast to the pattern in *A. dahraensis* isolated from eggplant). Aged clumps of conidia that developed on a single conidiophore were comprised of 40–50(68) conidia (Figure 7d). Secondary conidiophores were usually short and not the dominant element in the chain structure (except on water agar substrate); they were occasionally longer than the conidium body 14–80 × 3–7 (42 ± 16 × 4 ± 1) μm, with 2–8(10) transverse septa that mainly formed at the beginning of the chain (Figure 7e). These lateral or apical secondary conidiophores were single and had straight, geniculate or angular extensions. Juvenile conidia were ovoid and short-beaked with a tapered apex, or with a bluntly rounded base and apex containing 1–2(3) septa (Figure 7c). Mature conidia were narrowly oblong and ellipsoid with a rounded apex, pale brown or yellowish to dark brown, and the spore wall was either smooth or conspicuously ornamented (Figure 7a). Primary conidia at the base of the chain were predominately long-ellipsoid 28–75 × 6–18 (46 ± 12 × 12 ± 3) μm in size; with 5–7(9) transverse septa and at most one longitudinal septum in 1–3 of the transverse segments, which were slightly constricted near some transverse septa. Mature conidia at the beginning of the chain often produced secondary conidiophores (Figure 7b). The conidia in chains often showed a gradual decrease in size from relatively large and multi-cellular near the base of the chain to small and 1–3 celled near the apex. Most of the secondary conidia were the same length or shorter *ca* 29 ± 8 × 10 ± 3 (15–50 × 6–18) µm, with 1–4(8) transverse and 0–5 longitudinal septa. Other conidia were shorter *ca* 26 ± 6 × 9 ± 2 (17–42 × 6–15) µm and produced at least 3 transverse septa in the conidium body, terminating in a chain that was ovoid or ellipsoid and had no secondary conidiophores, beaks or conidiogenous loci on the apical cell.

Note: the most obvious difference between *A. humuli* and *A. pseudohumuli* was found in primary conidiophores; in *A. humuli,* they became variously branched upon aging and bore 8–10 conidiogenous sites [23], while in *A. pseudohumuli* they remained simple, geniculate, or sometimes produced two branches with only a few conidiogenous sites. However, secondary conidiophores of exceptional length within *A. humuli* conidial clusters were also present in *A. pseudohumuli* at a few positions but in most of the conidial clusters. Conidia size constitutes another distinction between the two species. *A. humuli* produced smaller, more mature conidia (28–38 × 10–11 µm) [23] than *A. pseudohumuli*.

*Alternaria cuminicola N. Bessadat*, and *P. Simoneau* sp. nov. (Figure 8)MycoBank: MB 835037Etymology: name refers to the closely related species *A. cumini* but with smaller conidia.Culture ex-type specimen: NB 65 = CBS 146567Other specimen examined: NB666, NB660, NB542, NB557

Description: the colonies on PCA were velvety, approximately 60 mm in diameter without concentric rings of growth and sporulated according to standardized conditions [23] (Figure 8f). At 7 days, sporulation occupied the center of the colony in a small area (1.5 cm diameter with NB65); however, many non-sporulating vegetative elements above the substrate surface were observed. Conidiophores that were produced in the central area were up to *ca* 370–770 × 3–5 µm, often developing further through geniculate extensions (Figure 8k). Development outside the center of the colony consisted of moderate primary conidiophores *ca* 106–350 × 4–5 µm (7–24 transverse septa), with few lateral branches, bearing a single apical conidium. Primary conidiophores were abundant and became densely crowded upon aging (up to 14 days) reaching a size length of 820–1100 × 4–5 µm (Figure 8j), with 3–6 lateral branches (22–190 × 3.5–5 µm). Each conidiogenous apex frequently bore a chain of 2–4(5) conidia, mainly with additional short lateral or basal chains of 2–3 conidia forming clumps of 12–40 conidia. Chain branching typically occurred by the elongation of secondary conidiophores (*ca* 15–50 × 3.5–5 µm with 2–5 transverse septa) from distal terminal conidial cells (Figure 8c), and lateral ones (Figure 8i) and subsequently formed numerous secondary, tertiary, and quaternary chains upon aging. Conidia that were produced on PCA were brown, short ellipsoid, ovoid, and beakless with a broad rounded or conical apex. The dominant size range was 20–29 µm long × 12–20 μm width with 2–4 transverse septa and 0–4 longitudinal septa. Conidia that were produced in the aerial branches tended to be small *ca* 6.5–25 × 5–18 µm with 0–3 transverse septa and 0–3 longitudinal septa (Figure 8b). Maturing conidia reached a maximum size of 30–38 × 12.5–20 μm with 2–5 transverse septa and 1–4 longitudinal septa (Figure 8a). They became distinctly constricted at the main two transverse septa and had a smooth or sometimes punctate surface. The outer walls and some transverse and longitudinal septa became thick and darkly pigmented.

Note: *A. cuminicola* sp. nov. isolates were separable at high magnifications from *A. cumini* reference strain CBS 121329. They clearly exhibited different sporulation patterns and conidial morphology under recommended conditions [23]. Conidia that were formed by *A. cuminicola* sp. nov. were smaller and the sporulation axes with conidia branching were more complex in structure, forming dense tufts that were visible to the naked eye in the center of the colony at 7–14 days.

### 3.3. Effect of Culture Media and Temperature on Colony Morphology and Growth

The effect of media and temperature on colony morphology and growth of the three newly described species was determined by the colony diameter method using PCA, PDA, MEA and OA medium. Growth was recorded at five temperatures (from 4 °C to 40 °C). Colony aspect and color varied with the type of culture medium. All fungi grew best on PCA and PDA at 25 °C (Table 2). In general, no growth was observed at 40 °C on any media except for NB530 (*A. dahraensis*) on PDA, MEA, OA and NB568 (*A. pseudohumuli*) on OA at 40 °C.

### 3.4. Pathogenicity Tests

Tomato plants were inoculated with 18 *Alternaria* isolates from the sections *Infectoriae*, *Ulocladioides*, *Eureka* and *Embellisioides*. Three isolates from the section *Alternaria*, i.e., *A. alternata* (isolates NB529 and NB553) and *A. arborescens* (NB555) were used as positive controls. However, regardless of the isolate origin or identity, similar necrotic symptoms were observed at 21 DAI on all inoculated plants, but with various intensities (Figure 9). No symptoms were seen in the negative control plants. The lesions initially developed on the basal leaves, the tips and along the margins of the leaf petiole. Symptoms also caused the development of brown, circular spots that turned into dark brown, necrotic spots surrounded by yellowish margins within 7 DAI. The infected lower leaves often began yellowing and browning before falling out. Lesions that were produced by aggressive isolates enlarged and coalesced, causing a blight of the leaves and progressing upwards. Two isolates from section *Infectoriae* (NB538, NB562) had a high severity on both tomato plants inducing severe blight and spot symptoms at 21 DAI (Figure 9A). The percentage of the leaf necrotic area on tomato plants was slightly higher with the reference isolates from section *Alternaria* (32.17 ± 9.53%) compared with those from section *Infectoriae* (16.67 ± 9.30%). Isolates belonging to sections *Eureka* and *Embellisioides* had the lowest severity (Figure 9B). Similar trends of aggressiveness were recorded for the potato plants, although in this case five strains from the section *Infectoriae* were found to be more aggressive than the two *A. alternata* strains that were used as positive controls (Appendix A).

To assess the host range of isolates from section *Eureka*, six species from *Apiaceae*, i.e., coriander, carrot, parsley, fennel, celery, cumin, were inoculated with five isolates (NB65, NB542, NB557, NB660 and NB666). Strains CBS 109383 (*A. petroselini*) and CBS 121329 (*A. cumini*) were used as positive controls. As expected, the most susceptible plant species were parsley with *A. petroselini* and cumin with *A. cumini* (Figure 10). The latter species produced symptoms on all tested plant species except for parsley and carrot. Symptoms were also observed on coriander, fennel and cumin plants with the five *A. cuminicola* sp. nov. isolates. Discrete symptoms were also observed on celery, except with NB666. However, these isolates were only weakly aggressive towards these plants, and non-pathogenic to parsley and carrot at 21 DAI. The inoculated leaves of cumin showed distinct blighted tips on lower leaves at 21 DAI, and sporulation was present on lesions after four weeks (Figure 10B).

## 4. Discussion

This study identified small-spored *Alternaria* spp. belonging to four sections other than those already described for isolates from symptomatic tissues of *Solanaceae* in north Algeria. Previous studies suggested a high diversity within the *Alternaria* component of the Algerian mycoflora associated with *Solanaceae.* At least five large-spored species from section *Porri,* i.e., *A. solani*, *A. linariae*, *A. protenta, A. grandis* and *A. crassa,* and two small-spored species from section *Alternaria* have been characterized from symptomatic tomato tissues [14,30,57]. Recently, *A. telliensis*, a new species from section *Japonicae* was also isolated from tomato [26]. The importance of the small-spored *Alternaria* species of section *Alternaria* in the development of foliar diseases in *Solanaceae* has been demonstrated in Algeria [14,18]. Small-spored *Alternaria* species were detected in about half of the sampled leaves with clear early blight and spot symptoms. Microscopic examinations showed that members of section *Alternaria* were dominant, representing more than 90% of the small-spored isolates, whereas species from other sections occurred at a lower frequency. Similar trends have been reported in the US [58] and Russia [24]. However, based on their morphology, nearly 8% of the small-spored isolates could not be considered to be members of section *Alternaria*. The present work focused on the diversity of these new small-spored *Alternaria* isolates that were associated with foliar lesions on *Solanaceae* in the north-western regions of the country. The combined use of morphological characteristics and phylogenetic analysis at the *gpd* locus revealed that these isolates could be classified into four different sections, i.e., sections *Infectoriae*, *Ulocladioides*, *Embellisioides* and *Eureka*. In agreement with other studies, although colony characteristics, such as color, texture, and sporulation patterns provided useful information for the preliminary separation of individuals into sections, their morphological characteristics were insufficient for accurate delineation at the species or even at the section level [21].

Two strains that were isolated from eggplant and black nightshade were assigned to the section *Ulocladioides*. This section has been introduced by Woudenberg et al. [46], who synonymized the genus *Ulocladium* with *Alternaria* and split the members of this genus into three sections based on *gpd-rpb2-tef1* multilocus phylogeny. This section is comprised of 21 recognized species [49], although sequence data for the taxonomy-informative loci are not available for all of them. The two isolates that were assigned to this section could not be differentiated on the basis of morphological characteristics. However, morphological characteristics of species formerly included in the genera *Ulocladium* are subject to variation according to growth conditions and can cause difficulty in establishing new taxa [48]. Members of this section were already described for tomato plants in Algeria and were identified as *A. consortialis* based on *ITS rDNA* sequence analysis [14]. This locus has limited value for distinguishing species within the *Alternaria* genus and this identification is therefore questionable. In fact, it was impossible to discriminate between the two isolates from section *Ulocladioides* that were under investigation in the present study and a group of phylogenetically related isolates using combined *gpd-rpb2-tef1* phylogeny. To better resolve the individuals within this section, Dang et al. [48] proposed the use of a *gpd-Alt a1-MAT 1-2-1-AGA1* phylogeny. Despite many attempts, we were unable to amplify at the latter locus. Sequence comparisons at the *gpd*, *Alt a1* and *MAT 1-2-1* were performed but they failed to discriminate the two isolates that were under study from two other phylogenetically related species (*A. cucurbitae* and *A. brassicae-pekinensis*).

More than half of the small-spored isolates from our strain collection were assigned to section *Infectoriae.* This section is characterized by the fact it includes species known to produce a sexual state by contrast to other *Alternaria* sections, along with additional characteristics (e.g., production of unique metabolites). A recent study even recommended the consideration of a taxonomic re-evaluation of the section *Infectoriae* which, due to the morphology, sexuality, genetic and mycotoxin profile of the species, could be defined as different fungal genus from *Alternaria* [59]. However, the *gpd* phylogeny that was used in our study to delineate sections confirmed that the section *Infectoriae* constitutes a well-defined group that is phylogenetically related to other groups defined within the genus *Alternaria* [46]. Most species within the section *Infectoriae* exhibited noticeable morphological similarities that could be resolved using phylogenetic analyses [52]. However, such analyses also revealed the high genetic variability of this section compared to others [59]. Using the combined datasets of *ITS rDNA, gpd, rpb2, tef1* and ATPase, new species have recently been defined within the section *Infectoriae* [51,55]. Using similar loci and comparisons with all of the species for which the corresponding sequences were available in databank, we showed that several isolates included in this section clustered independently with little to no statistical support for their phylogenetic position, which is similar to previous observations [54]. These authors recommended that species names should only be applied to isolates that possess a strongly supported phylogenetic position within the section *Infectoriae*. Five isolates from section *Infectoriae* could be clearly distinguished from each other by analyzing the above-mentioned set of sequences. They formed two well-supported groups, one at the upper part of the tree that we propose to include in a newly described species named *A. dahrensis* sp. nov., and another closely related but statistically different to *A. humuli* that formed a sister species named *A. pseudohumuli* sp. nov.

*Alternaria* species from section *Infectoriae* were detected as common species in food, the indoor environment and human cutaneous infections [23,51,60,61,62]. Recent studies have reported the incidence of species belonging to this section on potato plants in the Pacific North-west [58], in Pakistan [63], in Iran [17], in Russia [64], and our inoculation assays have indeed confirmed that the species from this section could be pathogenic to this host as well as to tomatoes.

The combined phylogenetic analysis from the *gpd*, *tef1* and *rpb2* genes indicated that isolates of the phylogenetically related sections *Eureka* and *Embellisioides* were isolated from *Solanaceae* in Algeria. A species belonging to the latter section (i.e., *A. chlamydosporigena*) was isolated recently from a tomato leaf spot in Iran [25]. The strain that was isolated from tomato leaves in Algeria (NB354) produced longer conidia, no chlamydospores, and grouped with *A. lolli*. Although the separation between NB354 and this species was statistically supported, as a single field isolate was available, we identified NB354 as a member of the species *A. lolii* (Woudenb. & Crous, Syn. *Embellisia lolii*). This species was described in *Lolium perenne* [56], a perennial ryegrass which was frequently observed in growing fields with many other weeds and wild *Solanaceae* such as black nightshade and jimson weed [28,57]. This may suggest that weeds may be an important source of *Alternaria* inoculum, contributing to the establishment of *Alternaria* spp. population in *Solanaceae* crops. Concerning the five members of section *Eureka*, all formed a well-supported group closely related to *A. cumini*, which has previously been reported on cumin in India [23] and Japan [65]. These isolates were considered as belonging to a new species based on morphological traits and multi-locus phylogenetic analysis and was circumscribed as *Alternaria cuminicola* sp. nov. Although all of these strains were isolated from tomato and black nightshade, they did not cause severe symptoms in the inoculated plants. Due to the phylogenetic relationship with *A. cumini*, the five isolates of *Alternaria*
*cuminicola* sp. nov. were inoculated onto six *Apiaceae* species, comparing their host range with other pathogenic species in this family. Our results showed that *A. cuminicola* isolates were able to infect coriander, fennel and cumin, albeit with low disease severity. This suggests that the preferred host of *A. cuminicola* probably belongs to another plant family as it was demonstrated recently for *A. telliensis*, which a species that was initially isolated in Algeria from *Solanaceae* and was recently found to be responsible for cabbage leaf spot in Iran [66]. Indeed, *Alternaria* spores are known as dry air spores that are well adapted for wind dispersal [67]. It is thus possible that this species has a broad host range and moves through the air from one crop to another. Areas with large fluctuations in humidity and temperature, and where agricultural activities are prevalent, may have the potential to significantly affect host–pathogen dynamics and epidemic development in natural plant–pathogen communities [68]. Northern Algeria, which is characterized by a Mediterranean climate and where cultural practices (potato and tomato fields in close proximity or even in rotation, solanaceous weed species often present near cultivated areas) may favor host jumps and the emergence of new virulence, and could therefore be considered as a potential hot spot of diversity for the genus *Alternaria.*

Our data indeed revealed a huge diversity of *Alternaria* spp. associated with leaf spot symptoms on *Solanaceae* in Algeria. In addition to the five large-spored species belonging to section *Porri* as well as members of the two species-complex *alternata* and *arborescens* from section *Alternaria* that have already been described in this country, here we documented the isolates belonging to four other sections and described three new small-spored *Alternaria* species whose role in the development of leaf spot disease has to be clarified. Indeed, small-spored *Altenaria* species are often considered as secondary plant pathogens that are able to either infect plants, together with large-spored *Alternaria* species, or to live on necrotic lesions caused by other plant pathogens. However, we previously showed the ability of strains of *A. alternata* and *A. arborescens* to provoke tomato and potato leaf blight and spot. These species may also develop synergistic interactions in mixed infections with moderately aggressive isolates of *A. linariae* [14]. The data reported here strongly suggest that some of the newly identified small-spored *Alternaria* species such as *A. dahraensis* sp. nov. and *A. pseudohumuli* sp. nov. might be able to infect tomato and/or potato plants and provoke significant leaf blight symptoms, at least under greenhouse conditions.

In conclusion, our results highlight the complexity of the etiology of early blight and brown spot of *Solanaceae* in Algeria and illustrate the necessity to better understand the pathogen population structure before devising a successful strategy of disease management. This might necessitate the characterization of additional phylogenetically informative molecular markers for the better delineation of species within some sections such as the sections *Infectoriae* and *Ulocladioides*.

## Figures and Tables

**Figure 1 life-11-01291-f001:**
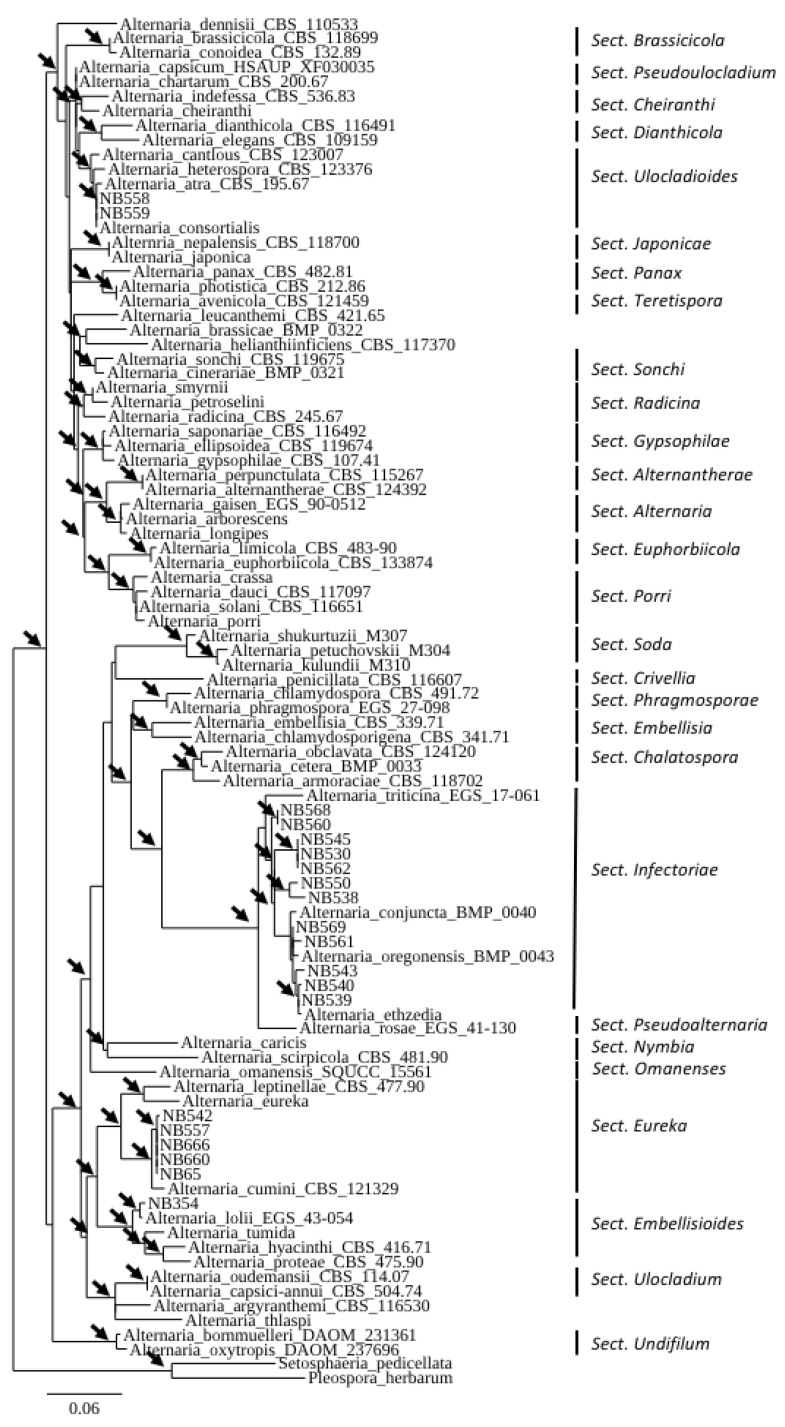
Phylogenetic tree reconstructed by the maximum likelihood method from the alignment of *gpd* sequences of 20 small-Scheme 75% are indicated by arrows. The GenBank acc. no. of sequences of the reference strains were from [46] except for members of the section *Soda* from [47] and section Omanenses from [22].

**Figure 2 life-11-01291-f002:**
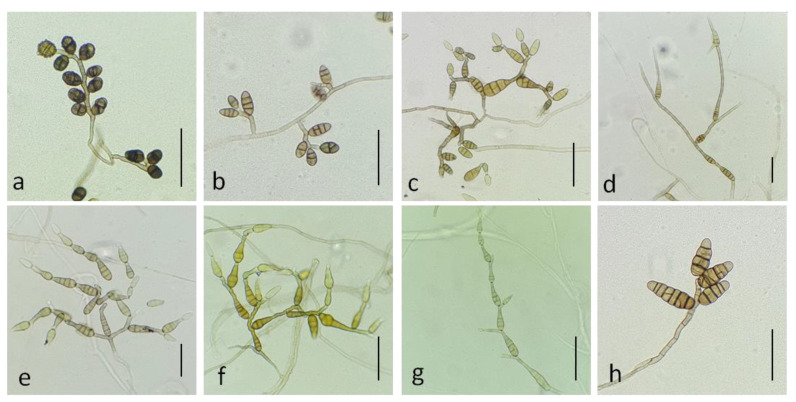
Conidia and conidiophores of *Alternaria* sp. sect. *Ulocladioides* NB559 (**a**), NB558 (**b**), *Alternaria* sp. sect. *Infectoriae:* NB545 (**c**), NB540 (**d**), NB569 (**e**), NB543 (**f**), NB561 (**g**), *Alternaria* sp. sect. *Embellisioides* NB354 (**h**). Bars = 50 µm.

**Figure 3 life-11-01291-f003:**
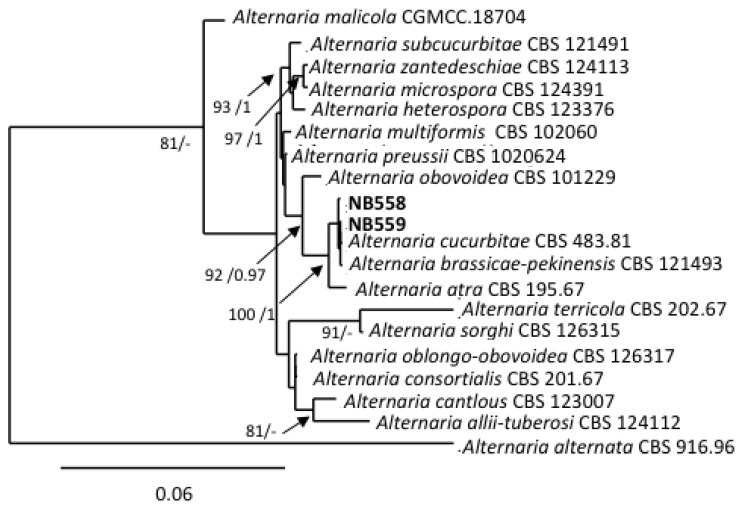
Phylogenetic tree reconstructed by the maximum likelihood method from the alignment of *gpd*, *Alt a1* and *MAT1-2-1* of *Alternaria* isolates from section *Ulocladioides*. Bootstrap support values greater than 75% and Bayesian posterior probabilities greater than 0.95 are indicated near nodes. The GenBank acc. no. of sequences of the strains included as references were retrieved from [48].

**Figure 4 life-11-01291-f004:**
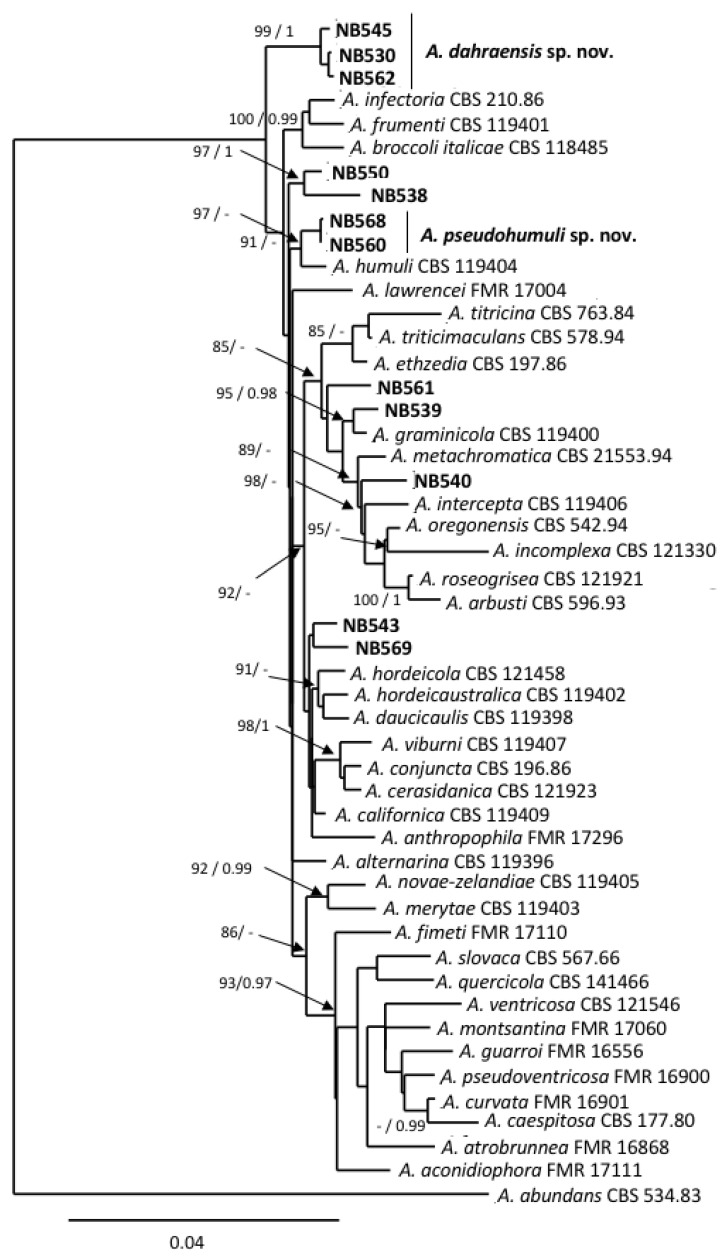
Phylogenetic tree reconstructed by the maximum likelihood method from the alignment of *ITS rDNA, gpd*, *rpb2*, *tef1* and *ATPase* of *Alternaria* isolates from section *Infectoriae*. Bootstrap support values greater than 75% and Bayesian posterior probabilities greater than 0.95 are indicated near nodes. The GenBank acc. no. of sequences of the strains included as references were retrieved from [51,55].

**Figure 5 life-11-01291-f005:**
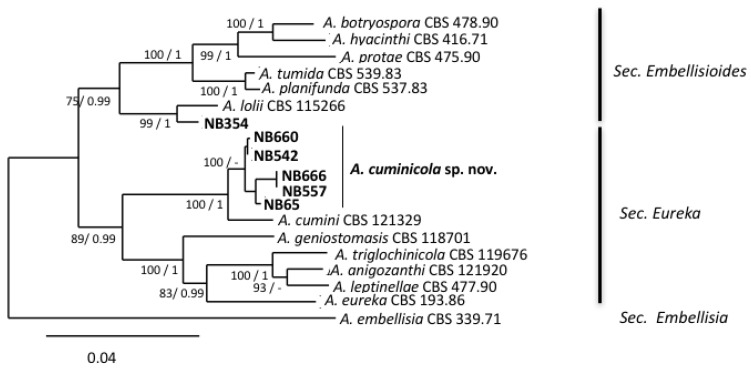
Phylogenetic tree reconstructed by the maximum likelihood method from the alignment of *gpd*, *rpb2*, *tef1* of *Alternaria* isolates from sections *Embellisioides* and *Eureka*. Bootstrap support values greater than 75% and Bayesian posterior probabilities greater than 0.95 are indicated near nodes. The GenBank acc. no. of sequences of the strains included as references were retrieved from [46].

**Figure 6 life-11-01291-f006:**
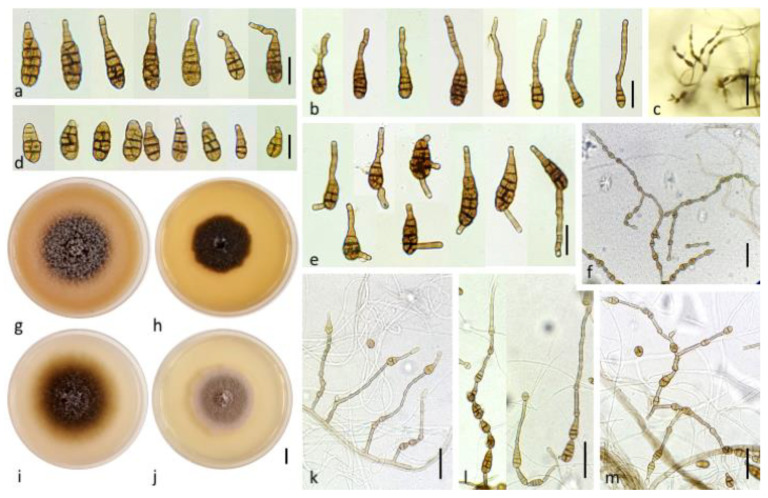
*Alternaria dahraensis* sp. nov. (**a**) Primary mature conidia. (**b**) Conidia with short and long apical secondary conidiophore (bars = 25 µm). (**c**) Short conidia chains produced on fertile aerial hyphae (bar = 100 µm). (**d**) Juvenile conidia with short-beak (right) and tapered apex (left), (bars = 25 µm). (**e**) Conidia with apical and lateral secondary conidiophores (bar = 25 µm). Colony aspect on: (**g**) Oatmeal Agar (OA), (**h**) Malt Extract Agar (MEA), (**i**) Potato carrot Agar (PCA), (**j**) Potato Dextrose Agar (PDA) after 7 days of incubation at 25 °C (bar = 10 mm). (**k**) Conidia and conidiophores produced in the central area of PCA colony (bar = 50 µm), (**l**) Conidiophores and conidia produced on periphery of PCA colony at 7–14 days (bars = 50 µm). Sporulation pattern on PCA at: (**m**) 14 and (**f**) 21 days (bar = 50 µm).

**Figure 7 life-11-01291-f007:**
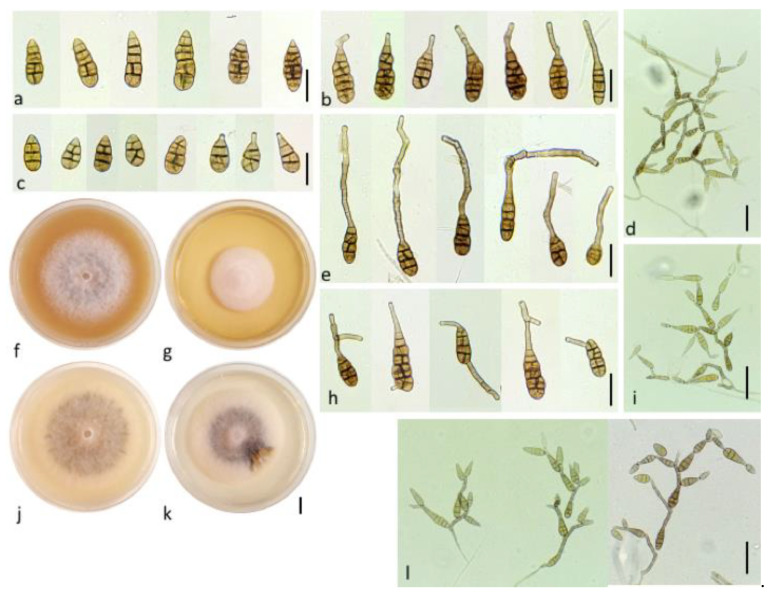
*Alternaria pseudohumuli* sp. nov. (**a**) Mature beakless conidia. Conidia with: (**b**) short and (**e**) long apical secondary conidiophore. (**c**) Juvenile conidia with short-beak (right) and tapered apex (left), (bars = 25 µm). Colony aspect on: (**f**) Oatmeal Agar (OA), (**g**) Malt Extract Agar (MEA), (**j**) Potato carrot Agar (PCA), (**k**) Potato Dextrose Agar (PDA) after 7 days of incubation at 25 °C (bar = 10 mm). (**h**) Conidia with lateral secondary conidiophores (bar = 25 µm). Sporulation pattern on water agar 2% at: (**l**) 7, (**i**) 14 and (**d**) 21 days, (bar = 50 µm).

**Figure 8 life-11-01291-f008:**
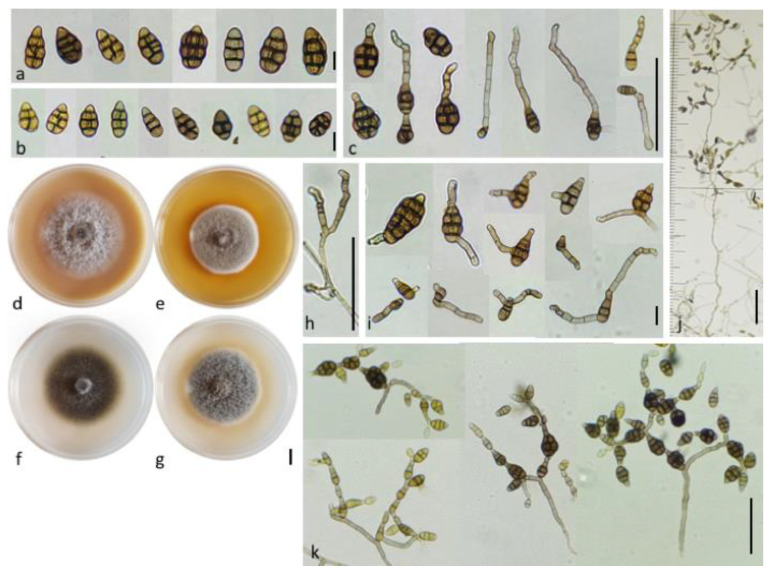
*Alternaria cuminicola* sp. nov. (**a**) Medium and mature conidia, (**b**) small conidia produced in aerial branches (bars = 10 µm), (**c**) conidia with one apical secondary conidiophore (bar = 50 µm). Colony aspect on: (**d**) Oatmeal Agar (OA), (**e**) Malt Extract Agar (MEA), (**f**) Potato carrot Agar (PCA), (**g**) Potato Dextrose Agar (PDA) after 7 days of incubation at 20 °C (bar = 10 mm). (**h**) Lateral branches of primary conidiophore (bar = 50 µm), (**i**) conidia with one or two lateral secondary conidiophores (bar = 10 µm), (**j**) Primary conidiophore with lateral branches and short conidia chains (bar = 100 µm), (**k**) sporulation pattern on PCA at 14 days (bar = 50 µm).

**Figure 9 life-11-01291-f009:**
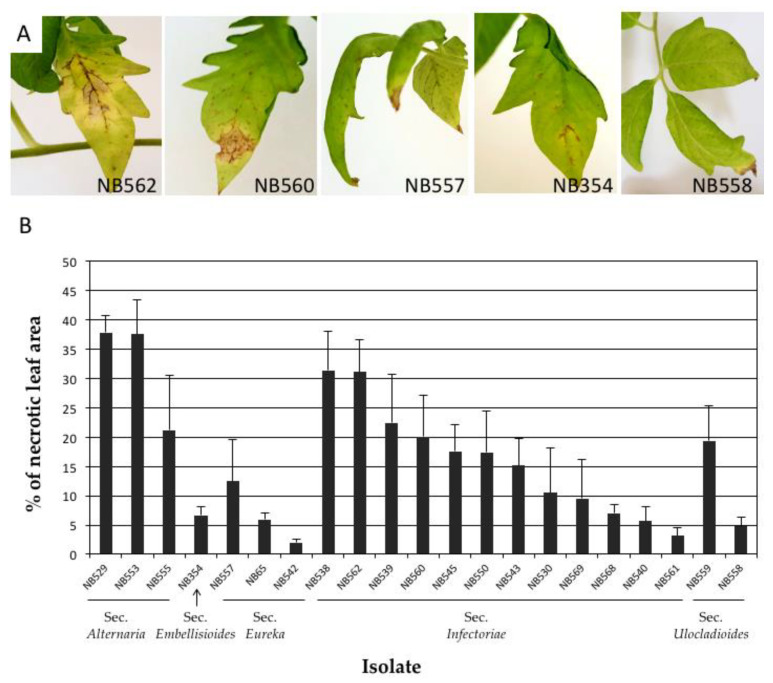
Pathogenicity testing of *Alternaria* isolates on tomato var. S^t^ Pierre. (**A**) Symptoms at 21 DAI on tomato leaves inoculated with *A. dahraensis* (NB562), *A. pseudohumuli* (NB560), *A. cuminicola* (NB 557), *A. lolii* (NB 354) and *Alternaria sp* section *Ulocladioides* (NB558); (**B**) Percentage of leaf necrotic area recorded at 21 DAI on plants inoculated with 18 *Alternaria* isolates plus 3 strains from section *Alternaria* (NB529, NB553, NB555).

**Figure 10 life-11-01291-f010:**
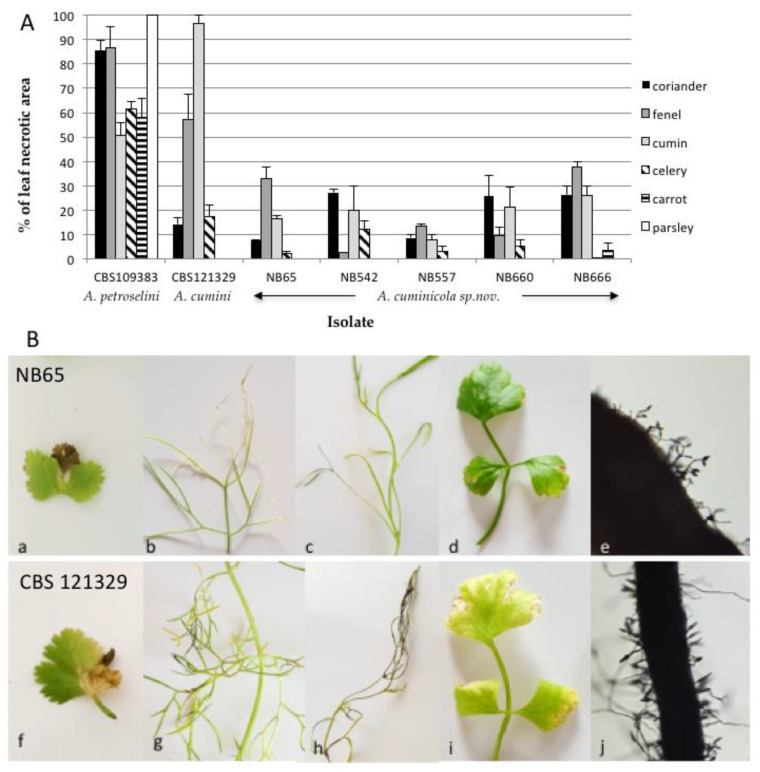
Pathogenicity testing of *Alternaria cuminicola* sp. nov. on *Apiaceae* plant species. (**A**) Percentage and mean of leaf necrotic area recorded at 21 DAI on *Apiaceae* leaves inoculated by isolates NB65, NB542, NB557, NB660 and NB666. The *Alternaria cumini* type strain CBS 121329 and *Alternaria petroselini* type strain CBS 109383 were included for comparison. (**B**) Symptoms of *Alternaria cuminicola* sp. nov. on *Apiaceae* at 21DAI. A Isolate NB65 on (**a**) coriander, (**b**) fennel, (**c**) cumin and (**d**) celery, (**e**) Spores of NB65 produced on cumin leaflet at 28 DAI. Isolate CBS 121329 (*A. cumini*) on (**f**) coriander, (**g**) fennel, (**h**) cumin and (**i**) celery, (**j**) Spores of CBS 121329 produced on cumin leaflet at 28 DAI.

**Table 1 life-11-01291-t001:** List of *Alternaria* spp. isolates selected for this study.

Isolate	Host	Symptoms ^b^	Section	GenBank Accession #						
				*ITS*	*gpd*	*ATPase*	*rpb2*	*tef1*	*Alt a1*	*Mat*
**NB65**	Tomato	BS	*Eureka*	OK353786	MK904515	MK913533	MK904529	MK904538	MK940316	/
NB354 ^a^	Tomato	BS	*Embellisioides*	OK353787	MK904520	/	MK904534	MK904543	/	/
**NB530** ^a^	Eggplant	BS	*Infectoriae*	OK353801	MK904502	MK913517	OK358873	OK358885	/	/
NB538 ^a^	Pepper	BS	*Infectoriae*	OK353802	MK904503	MK913518	OK358874	OK358886	/	/
NB539 ^a^	Pepper	BS	*Infectoriae*	OK353803	MK904509	MK913524	OK358875	OK358887	/	/
NB540 ^a^	Eggplant	BS	*Infectoriae*	OK353804	MK904510	MK913525	OK358876	OK358888	/	/
**NB542** ^a^	Black nightshade	BS	*Eureka*	OK353788	MK904516	OK358909	MK904530	MK904539	OK358902	/
NB543 ^a^	Eggplant	EB/BS	*Infectoriae*	OK353805	MK904513	MK913528	OK358877	OK358889	/	/
**NB545** ^a^	Potato	EB/BS	*Infectoriae*	OK353806	MK904504	MK913519	OK358878	OK358890	/	/
NB550 ^a^	Pepper	EB/BS	*Infectoriae*	OK353807	MK904505	MK913520	OK358879	OK358891	/	/
**NB557** ^a^	Tomato	BS	*Eureka*	OK353789	MK904517	OK358910	MK904531	MK904540	OK358903	/
NB558 ^a^	Black nightshade	BS	*Ulocladioides*	OK353790	MH232169	/	MK904526	MK904545	MN473190	OK358907
NB559 ^a^	Eggplant	BS	*Ulocladioides*	OK353791	MH287762	/	MK904527	MK904546	MN473191	OK358908
**NB560** ^a^	Eggplant	BS	*Infectoriae*	OK353808	MK904507	MK913522	OK358880	OK358892	/	/
NB561 ^a^	Potato	EB/BS	*Infectoriae*	OK353809	MK904512	MK913527	OK358881	OK358893	/	/
**NB562** ^a^	Tomato	EB/BS	*Infectoriae*	OK353810	MK904506	MK913521	OK358882	OK358894	/	/
**NB568** ^a^	Eggplant	BS	*Infectoriae*	OK353811	MK904508	MK913523	OK358883	OK358895	/	/
NB569 ^a^	Eggplant	EB/BS	*Infectoriae*	OK353812	MK904514	MK913529	OK358884	OK358896	/	/
**NB660**	Tomato	BS	*Eureka*	OK353792	MK904518	OK358911	MK904532	MK904541	OK358904	/
**NB666**	Tomato	BS	*Eureka*	OK353793	MK904519	OK358912	MK904533	MK904542	OK358905	/

^a^ Isolates selected for morphological characterizations; ^b^ BS = black spot, EB = early blight. Isolates identified as new species are indicated in bold characters. ^#^ Stands for Numbers.

**Table 2 life-11-01291-t002:** Cultural characteristics and temperature effect on new species growth after 7 days of incubation on PCA, PDA, MEA and OA.

					Colony Diameter (mm) at				
Species	Medium	Colony Type	Colony Color	Sporulation at 25 °C	4 °C	25 °C	30 °C	35 °C	40 °C
*A. cuminicola*	PCA	Velvety, appressed	Greyish green (30E6)	+++	5.4 ± 0.5	68.9 ± 1.7	54.3 ± 2.2	9.8 ± 1.0	-
	PDA	Cottony, dense	Greyish green (27E3)	+++	8.3 ± 0.3	65.6 ± 1.9	47.5 ± 1.1	7.4 ± 0.8	-
	OA	Velvety, fluffy	Dull green (30D4)	++	7.1 ± 0.3	64.6 ± 0.9	42.1 ± 0.9	5.5 ± 0.4	-
	MEA	Cottony, dense	greyish green (28ED5)	++	8.3 ± 0.5	55.0 ± 1.4	44.4 ± 1.1	6.4 ± 0.6	-
*A. dahraensis*	PCA	Arachnoid to loosely woolly	Olive brown (4E4)	+++	8.7 ± 1.7	76.1 ± 1.2	51.5 ± 1.0	9.5 ± 0.6	-
	PDA	Cottony compact	Olive (3D3) to orange grey (5B2)	++	6.5 ± 3.7	53.2 ± 2.2	51.8 ± 1.5	10.0 ± 0.4	5.3 ± 0.3
	OA	Cottony. fluffy	Olive (3E4) to greyish yellow (3C2)	++	8.1 ± 0.6	73.5 ± 1.0	60.3 ± 2.9	8.9 ± 1.0	5.4 ± 0.5
	MEA	Velvety	Olive (3E5)/(3E6)	+++	8.6 ± 0.8	50.0 ± 1.6	48.5 ± 0.6	9.8 ± 0.6	5.1 ± 0.3
*A. pseudohumuli*	PCA	Loosely woolly	Yellowish grey (4B2)	++	10.6 ± 0.5	79.4 ± 0.5	71.0 ± 0.8	13.0 ±1.8	-
	PDA	Cottony	Olive brown (4D3) center	+	10.5 ± 0.6	78.5 ± 0.7	54.6 ± 0.6	19.3 ± 1.0	-
	OA	Cottony. woolly	Greyish beige (4C2) center	+	11.8 ± 1.0	66.8 ± 0.4	74.0 ± 1.4	17.3 ± 1.0	5.3 ± 0.3
	MEA	Cottony dense	Olive brown (4D4) center	+	10.3 ± 1.0	67.6 ± 0.5	64.0 ± 0.8	13.1 ± 1.3	-

## Data Availability

All sequences obtained during this study have been deposited in GenBank and their accession numbers are provided. Specimen for newly described species have been deposited in the Westerdijk Fungal Biodiversity Institute (Utrecht, The Netherlands) in compliance with the Nagoya protocol. All isolates described in this study are freely available upon request to the authors.

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
