# Peer review of "Characterization of New Small-Spored *Alternaria* Species Isolated from *Solanaceae* in Algeria"

_life, 2021, doi:10.3390/life11121291_

Round 1
Reviewer 1 Report
The design of this study is similar to that of other modern works dedicated to biodiversity of plant pathogenic fungi. Representative collection of isolates was studied. Appropriate methods were used for all stages of research (morphology, phylogeny, and pathogenicity). Authors took into account the specificity of the object and selected culture media, conditions and genes that are usually used to study Alternaria fungi. Results are interesting and novel enough to be published in Life. The manuscript is well illustrated. Only a few corrections should be made.
- It is necessary to mark new species on the phylograms (figures 4 and 5).
- In would be very helpful for perception to add names of fungal species in figures 9B and 10A. They may be written as abbreviation with respective explication in the figure legend given below.
- Plant names mostly are given as common names, but in some cases Latin names appear instead of common. For instance, see lines 598 “black nightshade and Datura stramonium”, 493, 505-507 etc. Uniform style for plant names should be used. Preferable way is to use common names with Latin in parenthesis for first mentions.
- One sign of measurement unit “µm” is enough when length and width of conidia are given [e.g. (15)25-44(72) × (6)8-12(17) µm instead of (15)25-44(72) µm × (6)8-12(17) µm] in lines 250, 251, 282-285, 291.
- Line 97, title of the table 1. Strains of several species were studied therefore plural form should be used - Alternaria spp.
Author Response
We would like to thank the reviewers for their positive assessments on our manuscript and judicious advices. All modifications are highlighted in yellow in the revised version of the manuscript. We hope you will find this revised manuscript suitable for publication.
Answer to reviewer 1’s comments
- Concerning phylograms depicted on Fig 4 and 5 we have added the name on new species. We also modified Fig 9B, 10A and S3 to specify to which section belongs each isolate.
- In the text the common names of plant species are now used and the Latin names are provided in parenthesis for first mentions. We have deleted unnecessary ‘µm’ in lines 250, 251, 282-285, 291
- We modified the title of Table 1 as recommended by the reviewer.
Reviewer 2 Report
This manuscript is novel in terms of the information on Alternaria species isolated from Solanaceae. The authors offer important data to understand the characteristics of new small-spored Alternaria species. The manuscript is well-written to publish in this Journal, Life.

Author Response
We would like to thank the reviewers for their positive assessments on our manuscript and judicious advices. All modifications are highlighted in yellow in the revised version of the manuscript. We hope you will find this revised manuscript suitable for publication.
Answer to reviewer 2’s comments
- We slightly modified the structure of the introduction section in order to highlight the major problem and purpose of the study
- We have added one sentence at the beginning of the discussion section. However, we have the feeling that this section indeed discussed point by point the different results and addressed the main question posed, i.e. description of the diversity of Alternaria spp. associated to leaf symptoms of Solanaceae in Algeria and potential role of the different taxa in the etiology of the disease. We did not delete the paragraph related to isolates belonging to section Ulocladioides in the discussion section as proposed by the reviewer, as this is directly linked to the identification of two strains belonging to this section among our isolate collection.